# Advanced Signal Analysis Model for Internal Defect Mapping in Bridge Decks Using Impact-Echo Field Testing

**DOI:** 10.3390/s25216623

**Published:** 2025-10-28

**Authors:** Avishkar Lamsal, Biggyan Lamsal, Bum-Jun Kim, Suyun Paul Ham, Daeik Jang

**Affiliations:** Department of Civil Engineering, The University of Texas at Arlington, Nedderman Hall, 416 Yates St, Arlington, TX 76019, USA; axl0401@mavs.uta.edu (A.L.); bxl8392@mavs.uta.edu (B.L.); bxk6598@mavs.uta.edu (B.-J.K.)

**Keywords:** impact echo, deep learning, convolutional neural network, delamination

## Abstract

This study presents an advanced signal analysis model for internal defect identification in bridge decks using impact echo field testing data designed to mitigate signal noise and the variability encountered during real-world inspections. Field tests were conducted on a concrete bridge deck utilizing an automated inspection system, systematically capturing impact-echo signals across multiple scanning paths. The large volume of field-acquired data poses significant challenges, particularly in identifying defects and isolating clean signals and suppressing noise under variable environmental conditions. To enhance the accuracy of defect detection, a deep learning framework was designed to refine critical signal parameters, such as signal duration and the starting point in relation to the zero-crossing. A convolutional neural network (CNN)-based classification model was developed to categorize signals into delamination, non-delamination, and insignificant classes. Through systematic parameter tuning, optimal values of 1 ms signal duration and 0.1 ms starting time were identified, resulting in a classification accuracy of 88.8%. Laboratory test results were used to validate the signal behavior trends observed during the parameter optimization process. Comparison of defect maps generated before and after applying the CNN-optimized signal parameters revealed significant enhancements in detection accuracy. The findings highlight the effectiveness of integrating advanced signal analysis and deep learning techniques with impact-echo testing, offering a robust non-destructive evaluation approach for large-scaled infrastructures such as bridge deck condition assessment.

## 1. Introduction

Regular inspections of highway bridges are mandated by national regulations, with inspection intervals based on infrastructure type and classification [1,2,3]. Bridge decks, being directly exposed to traffic loads and environmental conditions, are particularly vulnerable to deterioration, leading to a shorter service life compared to other bridge components [4,5,6]. Therefore, monitoring delamination in bridge decks is essential for effective maintenance and rehabilitation strategies.

Structural health monitoring (SHM) and nondestructive testing (NDT) techniques have been extensively researched and applied over the past few decades [7,8,9,10,11]. Among the various NDT approaches, the impact-echo method has emerged as a highly effective technique for assessing bridge decks [12,13,14]. This method enables the detection of delamination by analyzing impact-echo signals, which exhibit distinct frequency characteristics in intact and defective areas. Several studies have demonstrated its effectiveness in detecting interface flaws, cracks, and voids in concrete structures, making it a widely adopted tool for infrastructure evaluation. Jiang et al. [15] successfully applied the impact-echo method to identify bonding interface flaws in ballastless track structures. They developed a flaw identification function that quantitatively classifies flaw types based on the amplitude ratio of two dominant spectral peaks, one corresponding to the bonding interface and the other to the bottom layer. Dawood et al. [16] successfully applied the impact-echo method combined with time-of-flight analysis to determine the depth of surface-opening cracks and employed shear-wave testing to generate three-dimensional visualizations of internal defects, such as voids in prestressed ducts and regions of corroded reinforcement.

Traditionally, impact-echo signal interpretation relied on manual analysis by professionals, which is time-consuming and prone to inconsistencies, particularly when processing large datasets. With advancements in automated robotic inspections, manual interpretation has become impractical for field applications [17,18,19]. Moreover, environmental factors and improper impactor alignment often introduce noise, leading to irrelevant or insignificant signals that distort delamination maps. To overcome these challenges, deep learning algorithms have been proposed for automated impact-echo signal analysis, significantly improving detection accuracy and efficiency [20,21,22].

Among deep learning techniques, convolutional neural networks (CNNs) have shown remarkable success in recognizing patterns within noisy datasets, making them effective for filtering irrelevant noise and enhancing the signal-to-noise ratio [23,24,25]. Recent studies have reported significant progress in applying CNN to analyze the NDT test results [26,27,28,29]. For example, Xu et al. [30] employed a CNN-based deep learning model to effectively analyze impact-echo signals, achieving an overall accuracy of 96.6% in detecting and classifying concrete structural defects; however, this performance was obtained using a relatively small dataset comprising only 168 samples from an experimental model. Similarly, Xiao et al. [31] applied CNN-based detection models to noisy X-ray images, improving defect identification and demonstrating superior robustness compared with conventional image-processing techniques, achieving a structural similarity index measure (SSIM) score of 0.70 for highly noisy defect images. Extending these applications, Virupakshappa et al. [32] utilized a CNN-based approach for multi-class defect classification in ultrasonic NDT signals, achieving an overall accuracy exceeding 90% in distinguishing flaw types and effectively addressing the limitations associated with manual inspection.

Despite the significant advancements in deep learning applications for nondestructive testing, the use of CNN for analyzing impact-echo field test results remains challenging [33]. The large volume of field-acquired data poses significant challenges, particularly in identifying defects and isolating clean signals and suppressing noise under variable environmental conditions. Existing approaches predominantly rely on fixed or empirically determined signal parameters, which often result in suboptimal noise reduction and inaccurate delamination mapping. A major limitation of these conventional methods is their inability to account for the inherent variability of field conditions, including environmental noise, incomplete wave responses, and impactor misalignment, all of which contribute to the generation of insignificant signals and misleading defect assessments [14]. In the context of impact-echo testing, insignificant signals refer to those with incomplete wave responses or low signal-to-noise ratios, typically arising from missing data, excessive noise, or poor surface conditions. These signals compromise the accuracy and reliability of defect detection by distorting delamination maps, thereby limiting the practical effectiveness of existing CNN-based classification techniques. Therefore, a thorough investigation into optimizing essential signal parameters—particularly starting time relative to the zero-crossing and signal duration—using CNN is crucial to improve the classification and mapping accuracy of impact-echo signals collected from field inspections. By systematically analyzing and optimizing these parameters, this study overcomes the limitations associated with conventional parameter selection and significantly improves delamination detection accuracy.

In this regard, the present study introduces two key innovations. First, it systematically optimizes critical signal parameters, specifically signal duration and starting time, for impact-echo field test data. Second, it enables precise classification of signals into delamination, non-delamination, and insignificant categories. A CNN–based classification framework, specifically tailored for field-acquired signals, is developed to minimize the influence of insignificant data and improve the reliability of delamination mapping. Unlike previous studies that applied CNNs in a general manner, the proposed approach uniquely integrates CNN-based classification with parameter optimization to enhance signal clarity, suppress environmental noise, and significantly improve defect detection accuracy. Comparative analyses of delamination maps before and after optimization demonstrate marked improvements in detection accuracy and mapping reliability. Thus, the present study provides a novel and practical framework for parameter-driven deep learning optimization, addressing key limitations of existing impact-echo methods and advancing the accuracy and robustness of non-destructive evaluation for bridge deck inspection.

## 2. Experimental Program

The difficulties and constraints of existing signal extraction techniques for advanced impact-echo data in field testing stem from varying environmental factors and the existence of multiple impact sources [34]. The application of deep learning algorithms has been proposed as a promising solution to address these issues. In this study, CNN technique using images is employed to process and categorize different types of signals. An overview of the proposed workflow is illustrated in Figure 1.

### 2.1. Experimental Details

The impact-echo test operates based on two distinct frequency modes: the thickness mode and the flexural vibration mode. The thickness mode occurs when body waves repeatedly reflect between parallel boundaries, while flexural mode is typically associated with thin delamination, characterized by out-of-plane vibrations above the delaminated area. In the laboratory test, the flexural vibration mode was specifically employed for delamination detection, which can be described as follows [35]:(1)$$f_{FM}=\frac{k_{DF}^{2}\pi }{2h^{2}}\sqrt{\frac{F_{r}}{\rho h}}$$
where $k_{DF}$, a dimensionless frequency, is derived from the natural frequencies in relation to the width-to-depth ratio of delamination. In Equation (1), h represents the depth of delamination, $F_{r}$ denotes flexural rigidity, and ρ indicates material density, respectively. The frequency of the flexural vibration mode is influenced by the geometric properties of the delamination such as its width and depth, which play a critical role in determining the corresponding frequency. Details of the impact-echo procedure can be found in the previous work [34]. In this study, a frequency window of 1–5 kHz was adopted, since it encompasses the dominant vibration modes of the delamination sizes considered. Accordingly, the energy intensity (EI) is expressed as:(2)$$EI_{n}\left(f\right)=\int_{f_{l}}^{f_{h}}\Psi_{n}\left(f\right)df=\sum_{i=1}^{N}\Psi_{n}\left(f_{i}\right),(f_{1}=f_{l},f_{N}=f_{h})$$
where $EI_{n}\left(f\right)$ and $\Psi_{n}(f)$ represent the nth MEMS sensor’s energy intensity and spectral energy density, respectively, ($\Psi_{n}\left(f\right)={\left|X_{n}\left(f\right)\right|}^{2}),\left|X_{n}\left(f\right)\right|$ is the magnitude of the frequency component, and $f_{l}\;$, and $f_{h}$ are the low and high-boundary frequencies (1 kHz and 5 kHz), respectively.

The details of the laboratory test calibration and production of the delamination map are described in the previous study by Kang et al. [34]. The testing slab measures 2.1 m×1.7 m and contains three artificial delamination at two different depths. Specifically, three thin plastic square plates—measuring 20 cm, 30 cm, and 46 cm—were embedded within the slab at depths of 10 cm and 4 cm, respectively, as illustrated in Figure 2. Delamination designated as D1 to D3 are located at a depth of 4 cm, whereas D4 to D6 are placed at a depth of 10 cm. An automated impactor system driven by a 12 V DC motor was employed to ensure a uniform and repeatable impact rate, following the configuration described in the previous literature [34]. The impact mechanism consists of a 12 mm low-carbon steel ball attached to a high-elastic stainless-steel wire, effectively minimizing undesired resonance frequency noise, as recommended by Kang et al. [36]. The resulting impact signals were captured using a non-contact MEMS sensor integrated with a data acquisition platform (USB-6366, National Instruments, Austin, TX, USA) and managed through LabVIEW software. Figure 3 displays the laboratory test results for delamination, illustrating the accumulated energy within the 1 kHz to 5 kHz frequency range. Regions with high energy, representing delamination, are marked in red, while areas with low energy, indicating no delamination, are shown in blue. The high-pixel clusters adjacent to D2–D4 can be attributed to several contributing factors: (i) wave interference and scattering at free boundaries or insert edges that concentrate flexural energy near the defect regions, (ii) process-induced secondary imperfections such as micro-voids or partial debonding generated during slab fabrication, (iii) local stiffness discontinuities, such as nearby reinforcement that induce band-limited resonant responses, and (iv) sampling or interpolation artifacts that may locally amplify these signal features.

An integrated bridge monitoring system referred to as automated crack evaluation (ACE) system was employed for high-speed field inspections on a 60 m long, three-lane-wide bridge. This advanced system integrated cutting-edge technologies, including double-sided bounce impactors (DSBI), multichannel acoustic scanning (MAS) units, and an automated height-adjustable scanning (AHAS) platform [34]. These features facilitated efficient detection of delamination through flexural vibration modes. The ACE system utilized 9 motorized impactors, and 22 MEMS sensors strategically distributed across the scanning platform, enabling high-resolution data acquisition. To ensure accurate measurements, motor operations were synchronized with controlled delays, effectively preventing signal overlap. The inspection covered an 8 m wide area, completed in three scanning paths. The system configuration is illustrated in Figure 4, with further technical details available in Kang et al. [34].

### 2.2. Deep Learning-Based Signal Extraction Algorithms

In this study, a CNN model is employed to analyze time-domain signals labeled according to damage types: delamination, non-delamination, and insignificant responses. These classifications are depicted in Figure 5. Figure 5 presents representative raw, unnormalized signals. The absolute amplitude differences arise from variations in energy partitioning among wave modes and local coupling conditions. In the delaminated region (b), flexural resonance above the defect traps vibrational energy within the 2–5 kHz range, resulting in higher low-frequency amplitudes and slower decay. In contrast, in the non-delaminated region (a), the energy propagates more efficiently into thickness and body-wave modes, producing higher-frequency components and faster attenuation, and consequently smaller amplitudes in the 1–5 kHz band. The insignificant response (c) exhibits a low signal-to-noise ratio (SNR) and incomplete wave cycles, leading to small amplitude and a spectrally diffuse signature.

An invalid signal is defined as one exhibiting incomplete wave motion. In the time domain, such a signal lacks full oscillation cycles and a clear attenuation trend. In the frequency domain, its peak frequency does not correspond to the expected geometric information and often appears as low-frequency noise in areas representing intact material. These signals are considered experimental artifacts arising from the data acquisition process rather than the intrinsic properties of the material. The primary cause is inadequate coupling between the sensors and the test surface, commonly encountered in field conditions such as measurements performed on transverse cross slopes or near bridge joints, where achieving firm and consistent sensor contact is challenging. To effectively distinguish between these categories, two key parameters are examined: the signal’s starting time ($S_{t_{i}}$) and its duration ($D_{t_{i}}$). The CNN is trained to determine the optimal $S_{t_{i}}$ and $D_{t_{i}}$ values that yield the highest classification accuracy. Once identified, these optimized parameters are integrated into the signal extraction process, significantly enhancing the precision of the resulting delamination maps. The CNN architecture consists of convolutional and pooling layers that process the input data hierarchically. The convolutional layers extract features by applying various filters to localized regions of the input, generating feature maps that capture the essential attributes of the signals. The pooling layers then reduce the dimensionality of these feature maps, preserving the most important information while minimizing computational demands. The full processing workflow is illustrated in Figure 6, showing how raw signals are progressively transformed into feature-rich representations that enable accurate damage classification. For the training process, 80% of the dataset was allocated for model learning, while the remaining 20% was retained for validation in order to optimize the damage identification framework. The detailed parameters of the proposed CNN are provided in Table 1.

The central concept of this approach involves constructing a specialized database that systematically records variations in signal parameters, such as starting time and duration. This parameter-focused dataset serves as the basis for distinguishing between meaningful signals, which are critical for producing accurate impact-echo evaluations, and insignificant signals, which could otherwise introduce noise and reduce the reliability of the assessment results. Figure 7 and Figure 8 illustrate the time-domain analysis range with $D_{t_{i}}$ and $S_{t_{i}}$. Figure 7 examines ten distinct durations ranging from 0.2 ms to 2 ms (e.g., $D_{t_{1}}=0.2\;ms$, $D_{t_{2}}=0.4\;ms$, and $D_{t_{i}}=i\times 0.2\;ms$). Each duration corresponds to a specific temporal window used for signal analysis. Similarly, the starting time shown in Figure 8 explores eight different scenarios (e.g., $S_{t_{1}}=0\;ms$, $S_{t_{2}}=0.1\;ms$, and $S_{t_{i}}=(i-1)\times 0.1\;ms$ after the zero-crossing point). The first scenario, $S_{t_{1}}$, is defined to start exactly at the zero-crossing point, corresponding to the initiation of the structural signal recording. The following scenarios, including $S_{t_{2}}$ and $S_{t_{3}}$, are offset incrementally by 0.1 ms, resulting in delays of 0.1 ms and 0.2 ms, respectively, from the zero-crossing reference point. As illustrated in Figure 7 and Figure 8, the complete time response of each impact-echo test was initially collected up to 4 ms in length at a sampling rate of 500 kHz, corresponding to 2000 samples per record. From this full signal, truncated segments were systematically generated by applying one of the ten duration settings ($D_{t_{i}}$) and one of the eight starting offsets ($S_{t_{i}}$). In the duration analysis, 300 complete responses were available. Each response produced ten truncated signals (one for each duration, with the starting point fixed at zero), yielding 300 × 10 = 3000 images. In the starting-time analysis, the same 300 complete responses were used, but the window length was fixed (1 ms) while the 8 different offsets were applied, resulting in 300 × 8 = 2400 images. This ensures that for each original response, multiple overlapping and non-overlapping temporal windows are considered, allowing the CNN to evaluate how both signal duration and onset position influence classification accuracy. This extensive dataset ensures a comprehensive evaluation of the impact of signal duration and starting time in order to investigate the accuracy of damage identification.

## 3. Results and Discussion

This section discusses the results from two primary aspects. Initially, time-domain signals are successfully classified into delamination, non-delamination, and insignificant categories by analyzing different combinations of signal durations and starting times relative to the zero-crossing point. Subsequently, the delamination maps generated through the impact-echo method are further refined by applying the optimized parameters identified through the CNN-based signal extraction. The incorporation of these optimal $D_{t_{i}}$ and $S_{t_{i}}$ values lead to notable improvements in the precision and clarity of the delamination mapping. For clarity, the results presented in Figure 5, Figure 6, Figure 7, Figure 8, Figure 9, Figure 10 and Figure 11 are based on the laboratory slab specimen with known artificial delaminations, which provides controlled ground truth for parameter calibration. The subsequent results in Figure 12, Figure 13, Figure 14 and Figure 15 are based on the 60 m bridge deck field test, which demonstrate the applicability of the optimized parameters under real inspection conditions. Laboratory results are used to validate the physical meaning of the optimized parameters, while field results highlight their robustness for practical deployment.

### 3.1. CNN-Based Signal Extraction Results

Figure 9 presents the confusion matrices representing the CNN model’s classification performance across different signal durations ($D_{t}$). Classification accuracy is indicated by the sum of correct predictions along the diagonal of each matrix. For the model trained with $D_{t_{1}}=0.2\;ms$ [Figure 9a], the overall accuracy is 75.5%, with class-wise accuracies of 32.2% for non-delamination, 24.4% for delamination, and 18.9% for insignificant signals. In comparison, $D_{t_{2}}=0.4\;ms$ [Figure 9b] yields an improved delamination detection rate of 25.6%—the highest among all durations—suggesting its effectiveness in detecting damage, which is critical for structural health monitoring. However, its overall accuracy is only marginally better at 76.7%, and it underperforms in the non-delamination and insignificant signal classes (31.1% and 20%, respectively). The model with $D_{t_{4}}=0.8\;ms$ [Figure 9c] shows further improvement in overall accuracy (77.7%), with a more balanced distribution of class-wise performance. The best performance, however, is observed with $D_{t_{5}}=1\;ms$ [Figure 9d], achieving the highest overall accuracy of 78.8%. While the delamination detection rate here (23.3%) is slightly lower than in case (b), it still remains competitively high. More importantly, the model also demonstrates improved performance in the other two classes—34.4% for non-delamination and 21.1% for insignificant signals—indicating that longer signal durations help the model generalize better across all signal types. This level of accuracy is comparable to, and in some cases exceeds, that reported in recent studies employing deep learning for impact-echo analysis—for example, Dorafshan and Azari [12], who achieved similar accuracy levels in bridge deck evaluations. In contrast, the present approach emphasizes pre-classification parameter optimization rather than relying on fixed-length signals, which enhances both robustness and overall performance. These findings underscore a fundamental trade-off: maximizing delamination detection [case (b)] versus attaining balanced and robust performance across all categories [case (d)]. Given that practical applications demand both sensitivities to damage and reliability in classifying undamaged and noisy signals, the model with $D_{t_{5}}=1\;ms$ offers the best overall utility. Thus, case (d) is selected not only for its highest total accuracy but also for its consistent and well-rounded class-wise performance, demonstrating that longer signal durations enhance the CNN model’s capability for stable and reliable signal identification in the impact echo dataset.

**Figure 9 sensors-25-06623-f009:**
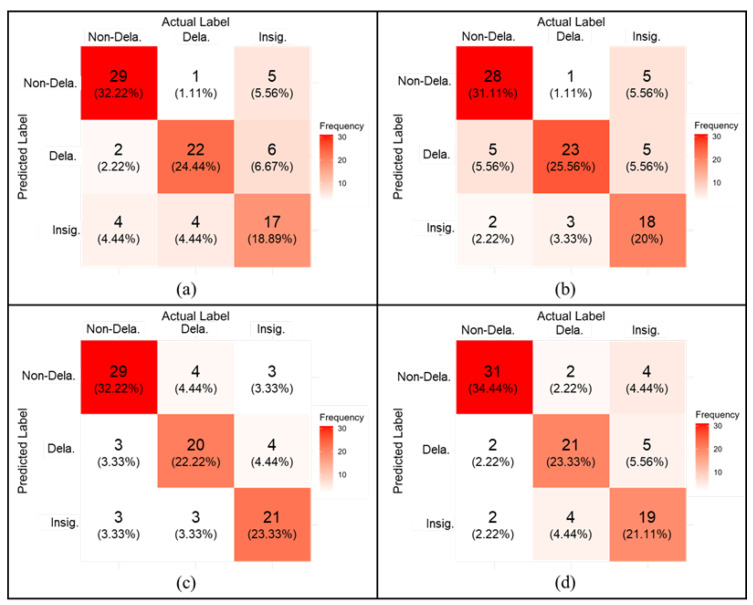
Confusion matrix of testing accuracy for different duration signals: (**a**) $D_{t_{1}}=0.2\;ms$, (**b**) $D_{t_{2}}=0.4\;ms$, (**c**) $D_{t_{4}}=0.8\;ms$, and (**d**) $D_{t_{5}}=1\;ms$.

Figure 10 illustrates the variation in testing accuracy as a function of different signal durations ($D_{t_{i}}$). The model reaches its peak classification accuracy at $D_{t_{5}}=1\;ms$. Although the overall variation across all ten durations is relatively small (all accuracies remain above 75% with a maximum difference of ~3%), this moderate improvement at 1.0 ms suggests that this duration provides a balanced window length that captures essential impact signals related to internal structural defects while avoiding excessive noise. This systematic tuning contrasts with conventional approaches in which signal duration is typically selected empirically. For example, studies by Xu et al. [30] and Wu et al. [33] successfully applied deep learning to impact-echo data but did not examine the sensitivity of model performance to signal duration gap that the present study directly addresses. Nevertheless, we acknowledge that the limited dataset size restricts the statistical strength of this conclusion, and that the “optimal” duration may vary with different batches of impact-echo signals, defect sizes, or deck geometries. As the duration increases to $D_{t_{6}}=1.2\;ms$, late-arriving components, including boundary reflections and residual resonance effects (highlighted in Figure 11), begin to enter the analysis window. These tail signals contribute weakly to delamination features but strongly to background variability, reducing model accuracy. When extended further to $D_{t_{8}}=1.6\;ms$, a more noticeable reduction in accuracy occurs, and by $D_{t_{10}}=2.0\;ms$, the accuracy declines considerably. This trend is primarily due to the inclusion of extraneous signal components, such as tail-end resonance and additional noise, which obscure the impact-related features and hinder precise classification. These results emphasize the necessity of optimizing signal duration to suppress irrelevant noise while preserving crucial information. Overall, a duration of $D_{t_{5}}=1.0\;ms$ offers the best compromise, capturing sufficient structural data while minimizing the effect of unwanted signal components. However, this finding should be viewed as an intermediate step rather than a universal rule; the optimal duration is likely influenced by structural parameters such as deck thickness, delamination depth, and defect size. Future studies with larger datasets and broader parameter coverage are needed to validate the generality of these trends.

**Figure 10 sensors-25-06623-f010:**
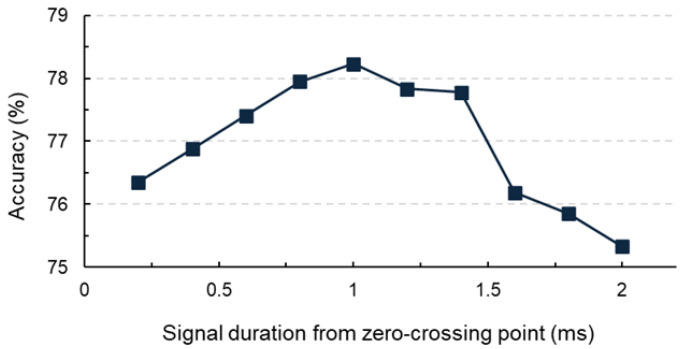
Plot of signal duration versus CNN accuracy, highlighting the optimal accuracy of 78.8% at a signal duration of 1 ms for accurate damage detection.

**Figure 11 sensors-25-06623-f011:**
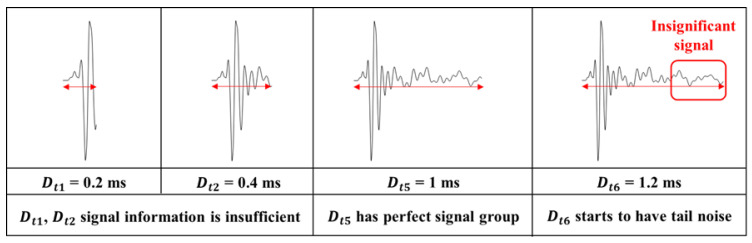
Impact of signal duration on damage detection accuracy highlighting insignificant signals due to wave reflections and overlap with adjacent impact events.

Figure 12 presents the classification results for different $S_{t_{i}}$ values across three selected $D_{t_{i}}$ durations. A consistent pattern emerges across all six scenarios, with the highest classification accuracy of 88.8% obtained at $S_{t_{2}}=0.1\;ms$, and similarly high accuracy observed at $S_{t_{3}}=0.2\;ms$. This improved performance is attributed to the elimination of the initial surface wave, which has a significant effect on signal characteristics. Surface waves (Rayleigh waves) are generated immediately after impact and propagate along the specimen’s surface at a velocity significantly lower than that of body waves (longitudinal or shear waves). However, since they travel directly along the surface between the impact point and the adjacent receiver, their path length is the shortest among all wave types. As a result, surface waves appear at the very beginning of the time-domain signal, preceding reflections from internal interfaces such as delaminations or the bottom boundary. The surface wave primarily occupies the first half-cycle of the signal, while the subsequent portions are dominated by mechanical echoes formed through the interaction of surface waves and P-wave reflections from delaminated areas or sound regions. In contrast, when $S_{t_{6}}=0.5\;ms$ is applied, accuracy declines as the signal predominantly contains tail-end noise and irrelevant components. Comparable trends are also observed in laboratory experiments, where the optimal defect identification occurs at $S_{t_{2}}=0.1\;ms$ or $S_{t_{3}}=0.2\;ms$. While other deep learning models, such as one-dimensional CNNs [25] and extreme learning machines [28], have been applied for structural damage detection, they typically require extensive pre-processing to filter out unwanted wave modes. In contrast, the present approach integrates this filtering step into the parameter optimization process, demonstrating that a carefully selected starting time can serve as a simple yet effective alternative to more complex signal-processing techniques. In the laboratory case, the availability of known defect locations provides a closer approximation to ground truth, whereas in the field, the labeling relied on flexural-mode energy intensity (EI), which correlates strongly with delamination but does not provide a perfect one-to-one match with actual defect boundaries. This approach may introduce labeling uncertainty; however, the agreement between lab-based ground truth validation and field EI-based labeling supports the reliability of the observed trends. Although the absolute accuracy values are slightly higher for the laboratory data (due to cleaner waveforms and reduced environmental noise), the consistent trend between field and laboratory results suggests that the optimized starting times are not artifacts of overfitting but reflect underlying wave propagation mechanisms. This consistency in classification trends between field and laboratory tests supports the reliability of the field accuracy evaluation, even in the absence of direct ground truth, and reinforces the physical validity of the optimized signal parameters.

**Figure 12 sensors-25-06623-f012:**
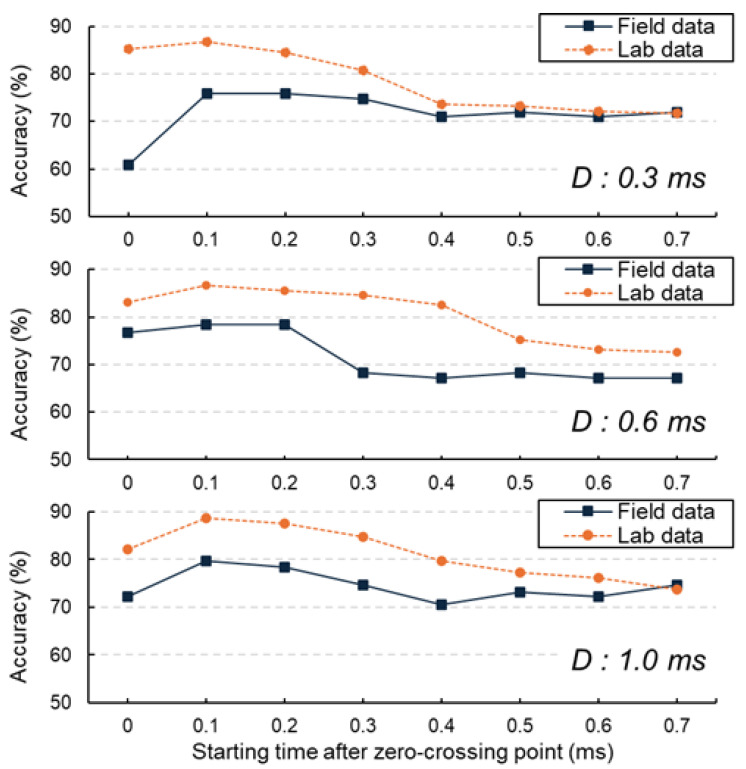
Accuracy comparison between field data and lab data at varying starting times after the zero-crossing point for signal durations 0.3 ms, 0.6 ms and 1.0 ms.

Figure 13 provides a detailed interpretation of the CNN accuracy results. In the scenario where $S_{t_{1}}=0\;ms$ and $D_{t_{1}}=1\;ms$, the initial segment of the signal primarily captures the pure surface wave, indicated by the red box. This component has minimal influence on the flexural mode of the impact-echo response. Since surface waves exhibit similar characteristics across all samples, they contribute little valuable information for CNN training. The P-wave, which is essential for identifying delaminations, arrives after this initial surface wave. When the starting times are adjusted to $S_{t_{2}}=0.1\;ms$ and $S_{t_{3}}=0.2\;ms$, the surface wave is effectively excluded, allowing the CNN to focus on a cleaner target signal composed of the impact-echo response combined with minimal surface wave effects and limited tail noise. This refinement results in higher-quality input data, improving the model’s ability to accurately detect damage. Conversely, as the starting time shifts to $S_{t_{4}}=0.3\;ms$, portions of the essential target signal are omitted, leading to reduced classification accuracy. This decline becomes more noticeable when a greater portion of the input signal consists of tail-end noise, as shown in the red box. These late-arriving components likely originate from secondary reflections at slab boundaries or from overlapping wave packets of adjacent impacts, both of which contribute spectral artifacts that diminish the discriminative power of CNN features. This analysis underscores the importance of selecting optimal starting times to maximize the utility of input signals for accurate classification.

**Figure 13 sensors-25-06623-f013:**
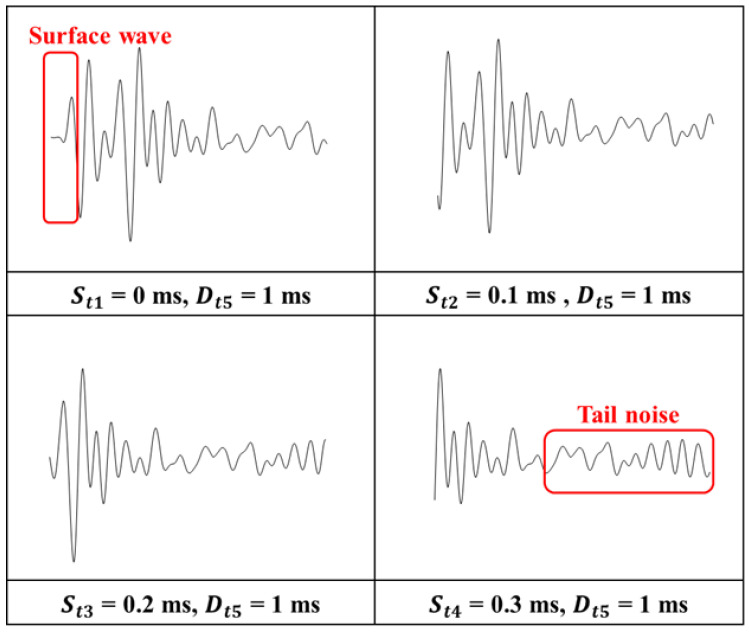
Illustration of the impact of surface wave and tail noise on signal quality for varying starting times with a fixed duration.

### 3.2. Delamination Results from Field Test

Delamination detection in field tests is frequently complicated by environmental disturbances and machine-induced noise, which can distort the acquired signals and hinder accurate interpretation [37,38,39]. In some cases, signals may be too weak to be captured, while in others, excessive amplification can result in misclassification of damage severity [24]. To overcome these obstacles, the present study systematically analyzed time-domain signals across a range of durations and starting times, aiming to minimize the influence of such interferences and improve detection reliability. The tested bridge deck has a total length of 60 m and a width of 13 m, with an average slab thickness of approximately 200 mm. Delamination results are presented for Lane 1, highlighted in red, as shown in Figure 14. This section provides an in-depth analysis of delamination results, as illustrated by the delamination maps comparing field test outcomes before and after incorporating CNN-optimized signal parameters. The maps demonstrate the impact of adjustments to starting times and signal durations on the accuracy of delamination detection. Although the optimized parameters ($D_{t_{5}}=1.0\;ms$ and $S_{t_{2}}=0.1\;ms$) proved robust in this case, it should be noted that deck thickness, reinforcement depth, wave velocity, and cover thickness can influence the precise arrival times of reflections and may shift the optimal values in other structures. Therefore, while the parameters identified here provide a practical reference framework, calibration to site-specific conditions remains important for field application. Subsequently, the observed differences are critically analyzed using signals at specific locations. Both time-domain and frequency-domain analyses are conducted to comprehensively assess the accuracy and reliability of the delamination maps, highlighting the performance improvements achieved through CNN-based signal optimization.

**Figure 14 sensors-25-06623-f014:**
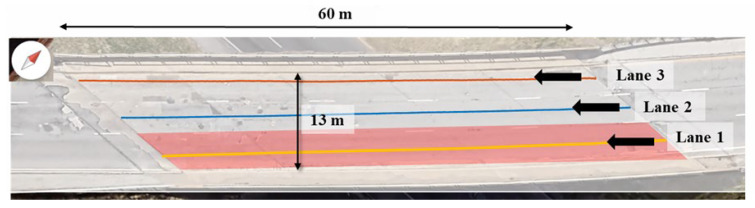
Scanned bridge deck highlighting lane 1 for delamination analysis.

Figure 15 presents the delamination maps obtained prior to and following the application of the CNN-optimized signal parameters. The original algorithm, depicted in Figure 15a, uses a 2 ms signal duration with no delay after the zero-crossing point. In contrast, Figure 15b applies CNN-recommended parameters, specifically a 1 ms signal duration with a 0.1 ms delay after the zero-crossing point. A detailed comparison of Figure 15a,b reveals noticeable differences: certain regions exhibit changes in mapped energy levels, with some points becoming more pronounced and others diminishing upon implementing the optimized signal parameters. The delamination maps obtained before and after applying the CNN-optimized signal parameters are displayed in Figure 15. In Region A, for example, the accumulated energy within the 1–5 kHz range in Figure 15a remains low, indicating no delamination. However, when processed using the CNN-optimized parameters ($D_{t_{5}}=1\;ms$ and $S_{t_{2}}=0.1\;ms$), a distinct low-frequency peak associated with delamination becomes evident. This improvement occurs because the extended signal duration in the original algorithm captures additional high-frequency noise, which suppresses the crucial low-frequency energy necessary for detecting delamination, as confirmed by both the time-domain and frequency-domain plots for this location. A similar pattern is observed in Region B. Region C further illustrates this effect. While Figure 15a classifies this area as free of delamination, Figure 15b indicates the presence of minor damage. The CNN-optimized analysis reveals a low-frequency peak characteristic of delamination, although its energy intensity is relatively weak. This variation may be caused by improper impactor contact due to surface irregularities such as potholes, which can cause the impactor to become stuck or produce unstable vibrations during impact. Region D offers another example: Figure 15a initially suggests a large, delaminated area, whereas Figure 15b, using the CNN-optimized parameters, identifies only minor damage. The corresponding signal in Region D exhibits a high-frequency dominant peak, indicating the absence of delamination. This outcome underscores the importance of proper impactor alignment and consistent energy transfer. Although higher impact energy can enhance wave reflection, energy magnitude alone is insufficient for accurate delamination detection; the frequency content and peak location within the spectrum are equally vital for correct interpretation. The higher spectral amplitudes in Figure 15b arise because the optimized 1 ms windows exclude trailing low-energy noise and concentrate the energy within the defect-sensitive frequency band. In contrast, the 2 ms windows in Figure 15a distribute the signal power over a longer duration, resulting in lower amplitude values in the frequency domain. Overall, these examples demonstrate that the CNN-optimized signal parameters significantly improve the accuracy of delamination mapping, as shown in Figure 15b, by minimizing the adverse effects of environmental noise and field conditions. This key finding directly addresses the issue of misclassification in field data reported by Lin et al. [14]. Rather than merely discarding invalid signals, the proposed method actively refines feature extraction across all signals, resulting in a more nuanced and reliable condition map that enhances the practical applicability of automated impact-echo systems in real-world scenarios. The improvements are not only visible in the maps but also supported by consistency with auxiliary inspections and by the physical explanation of spectral behavior. This optimized framework enables more reliable and precise damage assessments under real-world inspection scenarios.

**Figure 15 sensors-25-06623-f015:**
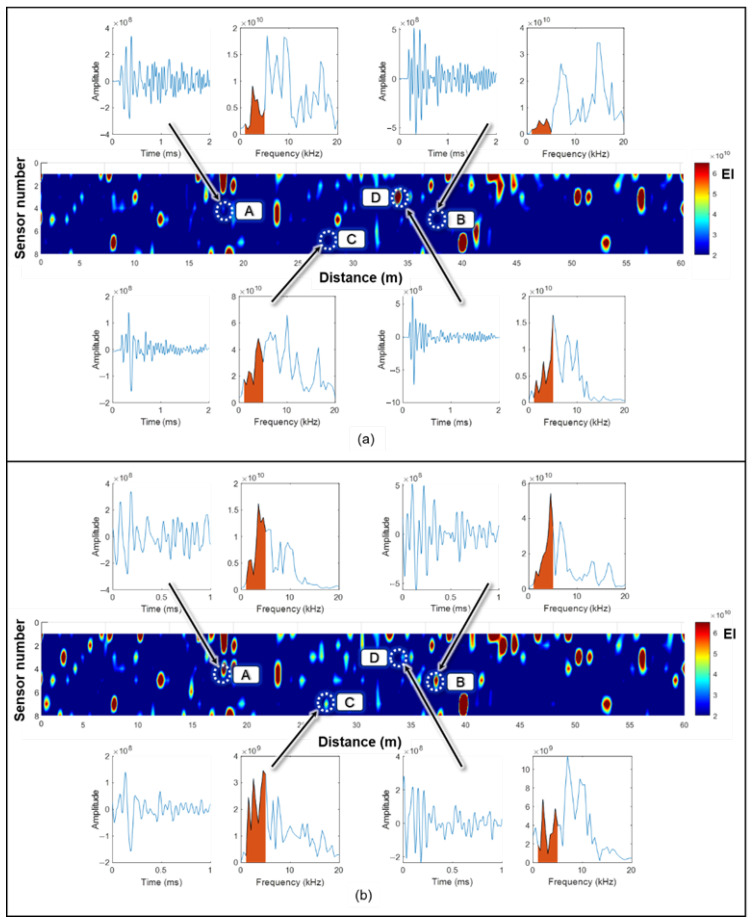
Delamination maps generated using (**a**) the original algorithm and (**b**) the CNN-optimized signal parameters from the signal extraction analysis (e.g., $D_{t_{5}}=1\;ms$ and $S_{t_{2}}=0.1\;ms$), accompanied by corresponding time-domain and frequency-domain signals for each case.

## 4. Concluding Remarks

This study presents a novel signal extraction framework that incorporates CNN to improve the accuracy of impact-echo testing, particularly by minimizing the effects of environmental variability often encountered in field data. Through the CNN-based optimization process, a signal duration of 1 ms and a starting time of 0.1 ms following the zero-crossing point were identified as the most effective parameters for this model. Integrating these optimized parameters into impact-echo delamination assessments led to substantial performance improvements, as verified through time-domain and frequency-domain analysis. This approach successfully mitigates issues related to signal loss and overestimation, which are commonly caused by inconsistencies in impactor energy and irregular ground conditions during field testing. The key outcomes of this research are summarized as follows:Deep learning analysis determined that a 1 ms signal duration combined with a 0.1 ms starting point after the zero-crossing delivers optimal damage detection performance, reaching an accuracy of approximately 88.8%.The CNN-based signal extraction method effectively reduces the misinterpretation of signals—both omissions and exaggerations—caused by environmental noise and field-testing variability.Implementing CNN-optimized signal parameters allows for more consistent and reliable interpretation of impact-echo results, particularly under challenging field conditions.Incorporating the optimized parameters into both time- and frequency-domain analyses significantly enhances the precision and reliability of damage detection.

## Figures and Tables

**Figure 1 sensors-25-06623-f001:**
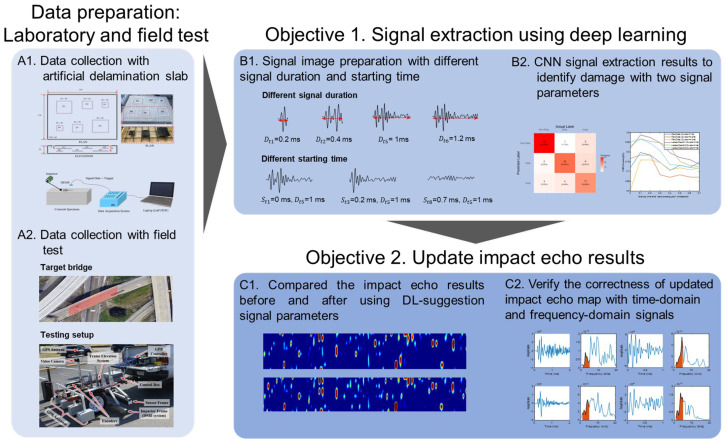
Flowchart of the proposed study.

**Figure 2 sensors-25-06623-f002:**
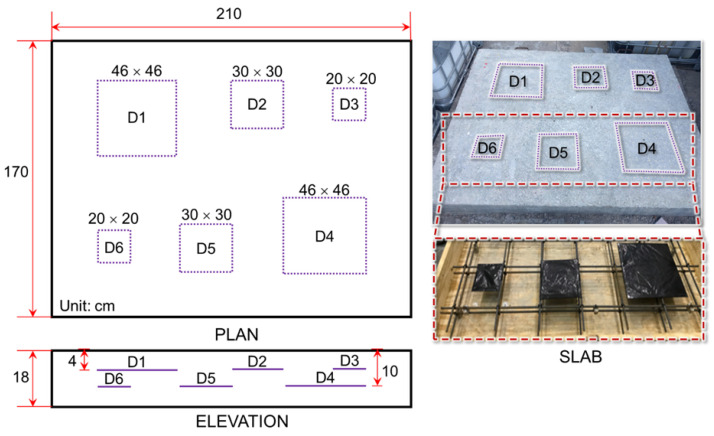
Testing slab with three different sizes and two different depts of delamination.

**Figure 3 sensors-25-06623-f003:**
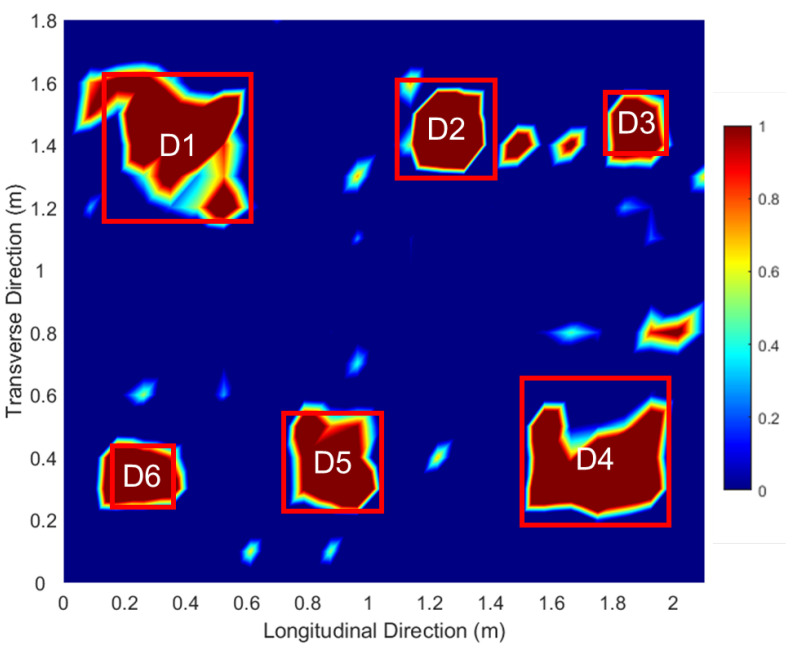
Two-dimensional interpolation map from the impact-echo laboratory testing.

**Figure 4 sensors-25-06623-f004:**
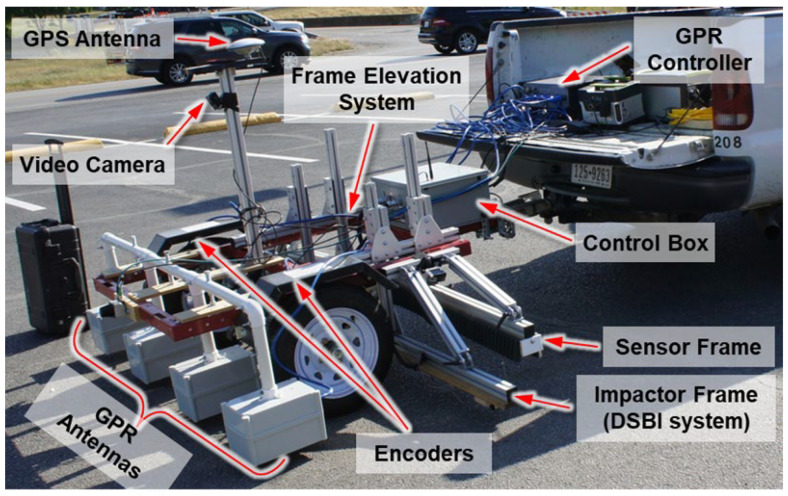
ACE system encompasses noncontact MEMS sensor, multi-channel impact-echo and GPR technologies.

**Figure 5 sensors-25-06623-f005:**
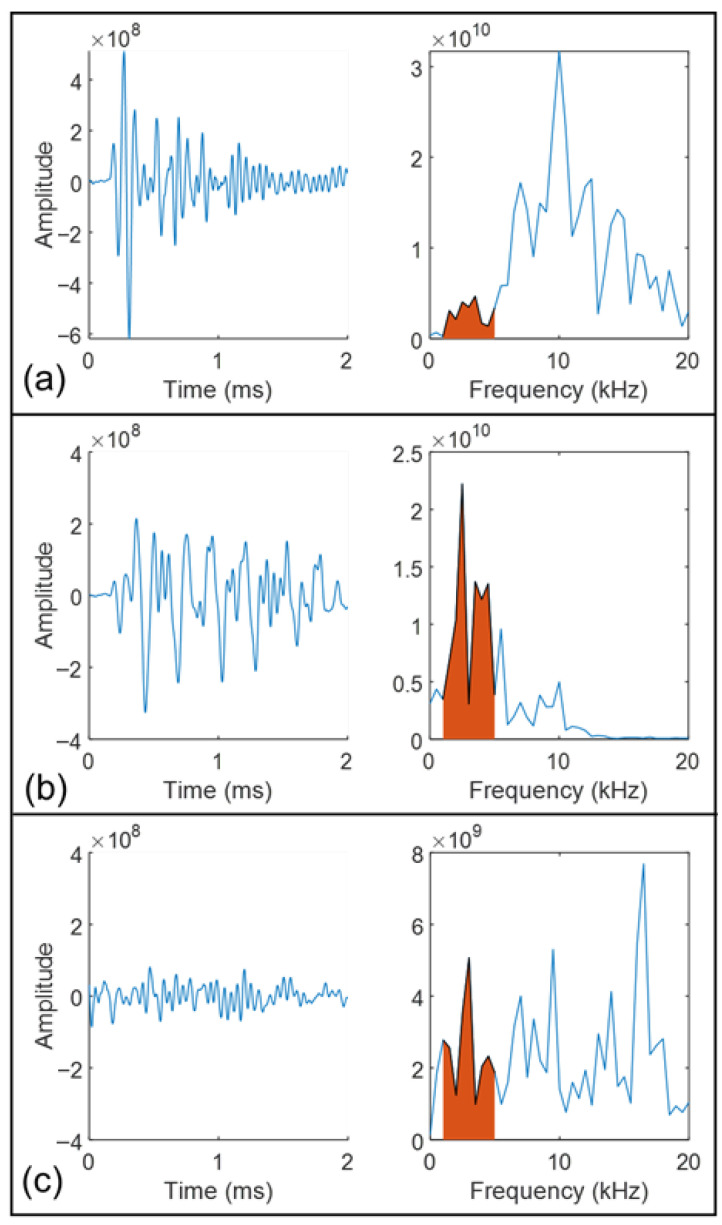
Representative time-domain and frequency-domain signals illustrating damage categories: (**a**) non-delamination, (**b**) delamination, and (**c**) insignificant signals. The red region illustrates the accumulated energy within the 1 kHz to 5 kHz frequency range.

**Figure 6 sensors-25-06623-f006:**
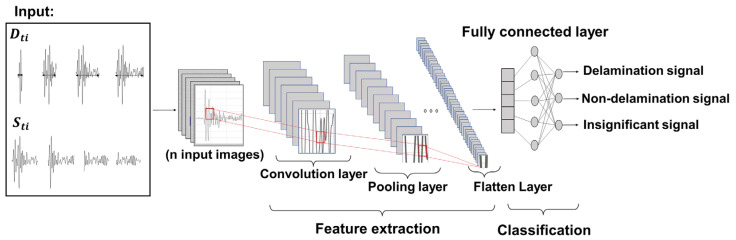
This CNN architecture used in the present study to analyze the input data.

**Figure 7 sensors-25-06623-f007:**
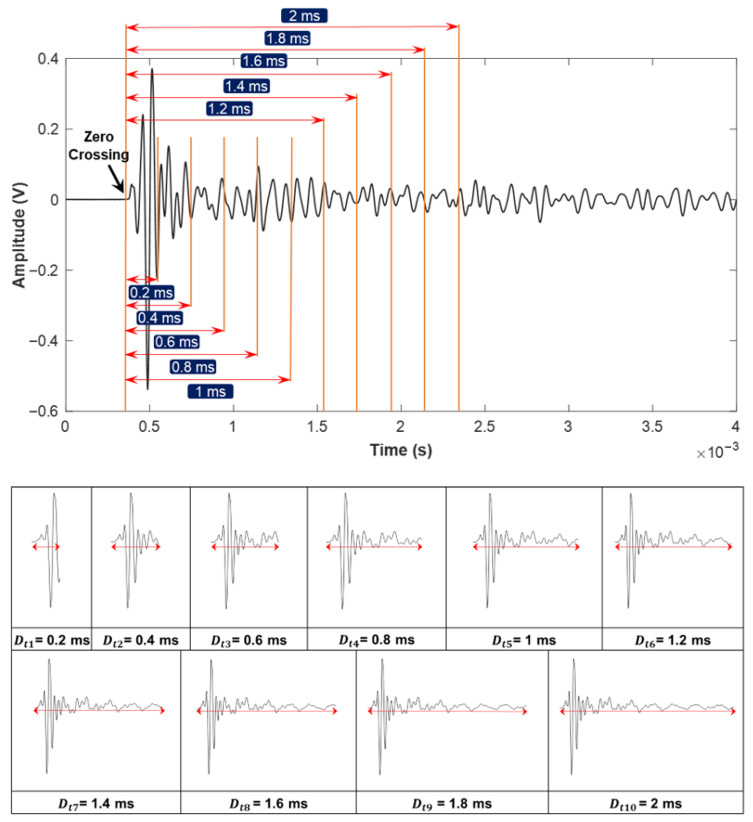
Range of signal durations (0.2 ms to 2 ms) applied as CNN input parameters for signal extraction.

**Figure 8 sensors-25-06623-f008:**
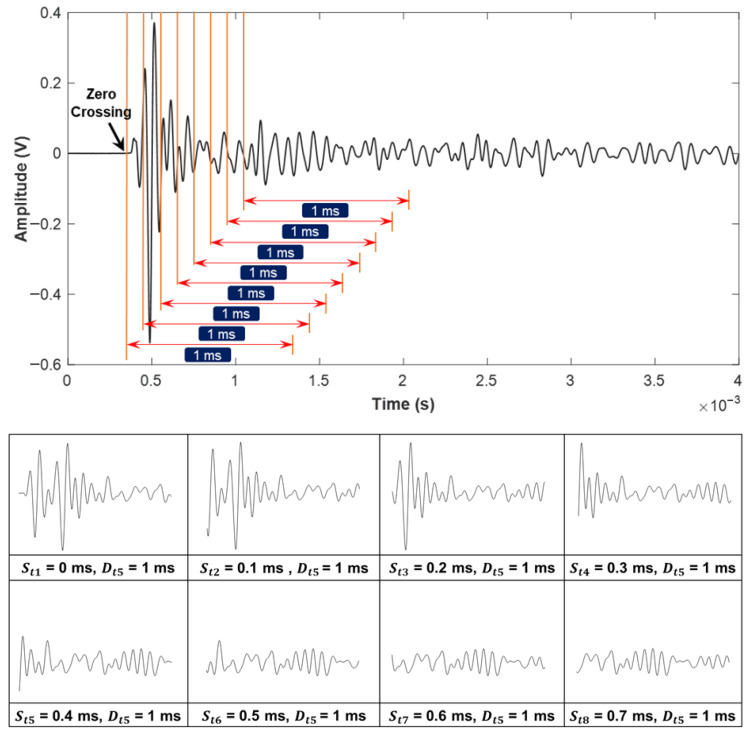
Range of starting times after the zero-crossing point, from immediate onset to 0.7 ms delay, utilized as CNN input parameters for signal extraction.

**Table 1 sensors-25-06623-t001:** Detail parameters of CNN algorithm used in this study.

No	Layer	Output Dimension	Filters	Kernel Size	Activation Function	Total Parameters
1	Input	200 × 200 × 200	–	–	–	–
2	Conv-1	196 × 196 × 32	32	5 × 5	ReLU	2432
3	Pool-1	65 × 65 × 32	–	3 × 3	–	0
4	Conv-2	61 × 61 × 64	64	5 × 5	ReLU	51,264
5	Pool-2	20 × 20 × 64	–	3 × 3	–	0
6	Conv-3	18 × 18 × 128	128	3 × 3	ReLU	73,856
7	Pool-3	6 × 6 × 128	–	3 × 3	–	0
8	Dense-1	512	–	–	ReLU	3,211,776
9	Dense-2	128	–	–	ReLU	65,664
10	Output	3	–	–	Softmax	387

## Data Availability

Data will be made available on request.

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
