# Peer review of "Advanced Signal Analysis Model for Internal Defect Mapping in Bridge Decks Using Impact-Echo Field Testing"

_sensors, 2025, doi:10.3390/s25216623_

Round 1
Reviewer 1 Report
Comments and Suggestions for Authors
The manuscript investigated the convolutional neural network (CNN)-optimized impact-echo field testing for interface defect mapping in bridge decks. The topic is interesting, and the paper is overall well-written. However, the following concerns should be addressed before the manuscript can be accepted for publication.
- The details of the impactor and the sensor implemented in the laboratory testing should be added. In addition, in Figure 3, can the authors explain why there exists some other high-pixel areas near D2, D3, and D4?
- It is easy to understand the signals corresponding to delamination and non-delamination cases. However, what do the authors mean “insignificant responses”? How do these responses correspond to experimental measurements?
- In Figure 5, can the authors explain why the signals have different amplitudes for the three cases?
- Can the authors expand a little more on the explanation of surface wave? Why will they be received at the very beginning of the signal?
- From the analysis, it seems the selected frequency range is very important. Did the authors try other frequency ranges? Like 1-3 kHz or 1-6 kHz? Some discussion on this aspect is also meaningful.
Author Response
Reviewer #1: The manuscript investigated the convolutional neural network (CNN)-optimized impact-echo field testing for interface defect mapping in bridge decks. The topic is interesting, and the paper is overall well-written. However, the following concerns should be addressed before the manuscript can be accepted for publication.
- The details of the impactor and the sensor implemented in the laboratory testing should be added. In addition, in Figure 3, can the authors explain why there exists some other high-pixel areas near D2, D3, and D4?
We appreciate the reviewer’s valuable comment. We agree that detailed information regarding the impactor and sensors used in the laboratory testing is essential. Accordingly, the relevant sentences have been newly added to the revised manuscript to provide this information, as shown below.
“An automated impactor system driven by a 12 V DC motor was employed to ensure a uniform and repeatable impact rate, following the configuration described in the previous literature [34[1]]. The impact mechanism consists of a 12 mm low-carbon steel ball attached to a high-elastic stainless-steel wire, effectively minimizing undesired resonance frequency noise, as recommended by Kang et al. [36[2]]. The resulting impact signals were captured using a non-contact MEMS sensor integrated with a data acquisition platform (USB-6366, National Instruments, USA) and managed through LabVIEW software.”
(Page 5, Line 153-160)
“The high-pixel clusters adjacent to D2–D4 can be attributed to several contributing factors: (i) wave interference and scattering at free boundaries or insert edges that concentrate flexural energy near the defect regions, (ii) process-induced secondary imperfections such as micro-voids or partial debonding generated during slab fabrication, (iii) local stiffness discontinuities, such as nearby reinforcement that induce band-limited resonant responses, and (iv) sampling or interpolation artifacts that may locally amplify these signal features.”
(Page 5, Line 163-169)
- It is easy to understand the signals corresponding to delamination and non-delamination cases. However, what do the authors mean “insignificant responses”? How do these responses correspond to experimental measurements?
Thank you for your valuable comment. The relevant sentences have been newly added to the revised manuscript to remove any ambiguities raised by the reviewer.
“An invalid signal is defined as one exhibiting incomplete wave motion. In the time domain, such a signal lacks full oscillation cycles and a clear attenuation trend. In the frequency domain, its peak frequency does not correspond to the expected geometric information and often appears as low-frequency noise in areas representing intact material. These signals are considered experimental artifacts arising from the data acquisition process rather than the intrinsic properties of the material. The primary cause is inadequate coupling between the sensors and the test surface, commonly encountered in field conditions such as measurements performed on transverse cross slopes or near bridge joints, where achieving firm and consistent sensor contact is challenging.”
(Page 6, Line 200-208)
- In Figure 5, can the authors explain why the signals have different amplitudes for the three cases?
We sincerely thank the reviewer for this valuable comment. The authors would like to clarify that the purpose of Figure 5 is to present representative raw, unnormalized signals to illustrate the physical differences in wave propagation behavior among sound, delaminated, and insignificant regions. As shown in Figure 5, absolute amplitude variations occur because the energy is partitioned differently among various wave modes and due to differences in local coupling conditions. In the delaminated region (Fig. 5b), flexural resonance above the defect traps vibrational energy within the 2–5 kHz frequency range, leading to higher low-frequency amplitudes and slower decay. Conversely, in the sound region (Fig. 5a), the energy propagates into thickness and body-wave modes, resulting in higher-frequency components and faster attenuation, thereby yielding smaller amplitudes in the 1–5 kHz band. The insignificant signal (Fig. 5c) shows a low signal-to-noise ratio (SNR) and incomplete wave cycles, leading to lower amplitude and more spectrally diffuse characteristics. These explanations have been clarified in the revised manuscript to ensure that the figure interpretation is better aligned with the underlying physical mechanisms.
To reflect the reviewer’s comment, the following sentences have been newly added to the relevant section.
“Figure 5 presents representative raw, unnormalized signals. The absolute amplitude differences arise from variations in energy partitioning among wave modes and local coupling conditions. In the delaminated region (b), flexural resonance above the defect traps vibrational energy within the 2–5 kHz range, resulting in higher low-frequency amplitudes and slower decay. In contrast, in the non-delaminated region (a), the energy propagates more efficiently into thickness and body-wave modes, producing higher-frequency components and faster attenuation, and consequently smaller amplitudes in the 1–5 kHz band. The insignificant response (c) exhibits a low signal-to-noise ratio (SNR) and incomplete wave cycles, leading to small amplitude and a spectrally diffuse signature.”
(Page 6, Line 190-199)
- Can the authors expand a little more on the explanation of surface wave? Why will they be received at the very beginning of the signal?
We appreciate the reviewer’s constructive comment. To clarify the explanation of wave propagation phenomena, additional descriptions have been included in the revised manuscript as follows
“Surface waves (Rayleigh waves) are generated immediately after impact and propagate along the specimen’s surface at a velocity significantly lower than that of body waves (longitudinal or shear waves). However, since they travel directly along the surface between the impact point and the adjacent receiver, their path length is the shortest among all wave types. As a result, surface waves appear at the very beginning of the time-domain signal, preceding reflections from internal interfaces such as delaminations or the bottom boundary.”
(Page 12, Line 354-360)
- From the analysis, it seems the selected frequency range is very important. Did the authors try other frequency ranges? Like 1-3 kHz or 1-6 kHz ? Some discussion on this aspect is also meaningful.
We sincerely appreciate the reviewer’s valuable comment. In this study, a frequency window of 1–5 kHz was adopted, and this range was not arbitrarily chosen. It was specifically selected because it was found to encompass the dominant vibration modes associated with the delamination sizes considered in our laboratory test slab. We acknowledge that other frequency ranges, such as the 1–6 kHz window mentioned by the reviewer, have also been effectively used in previous studies. The selection of an optimal frequency band generally depends on the specific objectives and structural characteristics of each investigation. For example, some studies[3] employ a broader range (1–6 kHz) to capture higher-order vibration modes or to improve sensitivity to smaller delaminations that resonate at higher frequencies. In our case, since the methodology was calibrated using a slab with known defect geometry, the 1–5 kHz frequency window was determined to be sufficient and reliable for capturing the flexural mode energy relevant to this study.
[1] Kang, S., Wu, Y. C., David, D. S., & Ham, S. (2022). Rapid damage assessment of concrete bridge deck leveraging an automated double-sided bounce system. Automation in Construction, 138, 104244.
[2] Kang, S., Ham, S., & Kim, K. J. (2021). An analytical, numerical, and experimental study of Rayleigh wave scattering for internal vertical crack evaluation. Construction and Building Materials, 306, 124838.
[3] Kang, S., Wu, Y. C., David, D. S., & Ham, S. (2022). Rapid damage assessment of concrete bridge deck leveraging an automated double-sided bounce system. Automation in Construction, 138, 104244.

Reviewer 2 Report
Comments and Suggestions for Authors
Reviewer’s report on “Advanced signal analysis model for internal defect mapping in bridge decks using impact-echo field testing”
The paper proposes a signal analysis model for detecting internal defects in concrete bridge decks using impact-echo signal data collected by an automated inspection system. The authors used a deep learning approach based on a convolutional neural network (CNN) to identify critical signal parameters and enhance the accuracy of defect detection.
Overall, the paper presents an interesting topic in the field of structural health monitoring. The real-world case study demonstrated the applicability of the proposed method. However, there are several points that the authors should consider before the paper can proceed further:
The introduction section requires substantial revision. The research problem should be stated more clearly, and the challenges associated with automated inspection of roadway infrastructure, including bridges, need to be better outlined. The literature review is not comprehensive; it currently lists several studies without discussing their key findings, limitations, or how they relate to the current work. Numerous studies have been published on the use of autonomous inspection systems and AI/ML techniques (including deep learning) for data analysis in similar contexts. The authors should clearly explain how this work differs from existing studies and what specific novelty it offers in terms of the defect detection, deep learning approach, or methodological advancement.
The experimental section is generally interesting; however, its organization requires some restructuring to improve clarity and logical flow. Some of the fundamental methodological details—such as Equations (1) and (2)—would be more appropriately placed earlier in a dedicated Methods section to provide readers with a clearer understanding of the analytical foundation before the experiments are described. On the other side, some parts of the experimental section currently include results that would be better positioned in the subsequent Results section.
The results themselves are presented clearly and appear to support the study’s objectives. However, the discussion of these results needs to be strengthened. In particular, the authors should provide a more critical interpretation of their findings, including comparisons with other state-of-the-art deep learning techniques or similar studies in the literature.
Author Response
Reviewer #2: The paper proposes a signal analysis model for detecting internal defects in concrete bridge decks using impact-echo signal data collected by an automated inspection system. The authors used a deep learning approach based on a convolutional neural network (CNN) to identify critical signal parameters and enhance the accuracy of defect detection. Overall, the paper presents an interesting topic in the field of structural health monitoring. The real-world case study demonstrated the applicability of the proposed method. However, there are several points that the authors should consider before the paper can proceed further:
- The introduction section requires substantial revision. The research problem should be stated more clearly, and the challenges associated with automated inspection of roadway infrastructure, including bridges, need to be better outlined. The literature review is not comprehensive; it currently lists several studies without discussing their key findings, limitations, or how they relate to the current work. Numerous studies have been published on the use of autonomous inspection systems and AI/ML techniques (including deep learning) for data analysis in similar contexts. The authors should clearly explain how this work differs from existing studies and what specific novelty it offers in terms of the defect detection, deep learning approach, or methodological advancement.
We sincerely appreciate the reviewer’s valuable comments. In response, a more in-depth literature review has been added to the revised manuscript to reflect the reviewer’s suggestions and strengthen the contextual background of the study. Additionally, the novelty and contribution of the present study have been further emphasized in the Introduction section of the revised manuscript.
“Jiang et al. [16] successfully applied the impact-echo method to identify bonding interface flaws in ballastless track structures. They developed a flaw identification function that quantitatively classifies flaw types based on the amplitude ratio of two dominant spectral peaks; one corresponding to the bonding interface and the other to the bottom layer. Dawood et al. [17] successfully applied the impact-echo method combined with time-of-flight analysis to determine the depth of surface-opening cracks and employed shear-wave testing to generate three-dimensional visualizations of internal defects, such as voids in prestressed ducts and regions of corroded reinforcement.”
(Page 2, Line 45-53)
“For example, Xu et al. [30] employed a CNN-based deep learning model to effectively analyze impact-echo signals, achieving an overall accuracy of 96.6% in detecting and classifying concrete structural defects; however, this performance was obtained using a relatively small dataset comprising only 168 samples from an experimental model. Similarly, Xiao et al. [31] applied CNN-based detection models to noisy X-ray images, improving defect identification and demonstrating superior robustness compared with conventional image-processing techniques, achieving a structural similarity index measure (SSIM) score of 0.70 for highly noisy defect images. Extending these applications, Virupakshappa et al. [32] utilized a CNN-based approach for multi-class defect classification in ultrasonic NDT signals, achieving an overall accuracy exceeding 90% in distinguishing flaw types and effectively addressing the limitations associated with manual inspection.”
(Page 2, Line 66-77)
“In this regard, the present study introduces two key innovations. First, it systematically optimizes critical signal parameters, specifically signal duration and starting time, for impact-echo field test data. Second, it enables precise classification of signals into de-lamination, non-delamination, and insignificant categories. A CNN–based classification framework, specifically tailored for field-acquired signals, is developed to minimize the influence of insignificant data and improve the reliability of delamination mapping. Un-like previous studies that applied CNNs in a general manner, the proposed approach uniquely integrates CNN-based classification with parameter optimization to enhance signal clarity, suppress environmental noise, and significantly improve defect detection accuracy. Comparative analyses of delamination maps before and after optimization demonstrate marked improvements in detection accuracy and mapping reliability. Thus, the present study provides a novel and practical framework for parameter-driven deep learning optimization, addressing key limitations of existing impact-echo methods and advancing the accuracy and robustness of non-destructive evaluation for bridge deck inspection.”
(Page 3, Line 99-113)
- The experimental section is generally interesting; however, its organization requires some restructuring to improve clarity and logical flow. Some of the fundamental methodological details—such as Equations (1) and (2)—would be more appropriately placed earlier in a dedicated Methods section to provide readers with a clearer understanding of the analytical foundation before the experiments are described. On the other side, some parts of the experimental section currently include results that would be better positioned in the subsequent Results section.
We sincerely appreciate the reviewer’s valuable comment. In response, the fundamental methodological details, including Equations (1) and (2), have been repositioned earlier in the manuscript to provide readers with a clearer understanding of the analytical foundation prior to the description of the experimental procedures. Furthermore, the manuscript has been thoroughly reviewed and revised to enhance overall clarity and logical flow.
Please refer to Section 2.1 (Experimental Details) in the revised manuscript for the corresponding modifications.
- The results themselves are presented clearly and appear to support the study’s objectives. However, the discussion of these results needs to be strengthened. In particular, the authors should provide a more critical interpretation of their findings, including comparisons with other state-of-the-art deep learning techniques or similar studies in literature.
We sincerely appreciate the reviewer’s valuable comment. We agreed that an in-depth discussion on experimental results is required. Accordingly, the following sentences have been newly added to the relevant section in the revised manuscript to reflect the reviewer’s comment.
“This level of accuracy is comparable to, and in some cases exceeds, that reported in recent studies employing deep learning for impact-echo analysis—for example, Dorafshan and Azari [13[1]], who achieved similar accuracy levels in bridge deck evaluations. In contrast, the present approach emphasizes pre-classification parameter optimization rather than relying on fixed-length signals, which enhances both robustness and overall performance.”
(Page 10, Line 300-304)
“This systematic tuning contrasts with conventional approaches in which signal duration is typically selected empirically. For example, studies by Xu et al. [30[2]] and Wu et al. [33[3]] successfully applied deep learning to impact-echo data but did not examine the sensitivity of model performance to signal duration gap that the present study directly addresses.”
(Page 11, Line 321-325)
“While other deep learning models, such as one-dimensional CNNs [25[4]] and extreme learning machines [28[5]], have been applied for structural damage detection, they typically require extensive pre-processing to filter out unwanted wave modes. In contrast, the present approach integrates this filtering step into the parameter optimization process, demonstrating that a carefully selected starting time can serve as a simple yet effective alternative to more complex signal-processing techniques.”
(Page 13, Line 367-372)
“This key finding directly addresses the issue of misclassification in field data reported by Lin et al. [15[6]]. Rather than merely discarding invalid signals, the proposed method actively refines feature extraction across all signals, resulting in a more nuanced and reliable condition map that enhances the practical applicability of automated impact-echo systems in real-world scenarios.”
[1] Dorafshan, S.; Azari, H. Evaluation of Bridge Decks with Overlays Using Impact Echo, a Deep Learning Approach. Autom. Constr. 2020, 113, 103133, doi:10.1016/j.autcon.2020.103133.
[2] Xu, J.; Yu, X. Detection of Concrete Structural Defects Using Impact Echo Based on Deep Networks. J. Test. Eval. 2021, 49, 109–120, doi:10.1520/JTE20190801.
[3] Wu, Y.; Zhang, J.; Gao, C.; Xu, J. Internal Defect Detection Quantification and Three-Dimensional Localization Based on Impact Echo and Classification Learning Model. Measurement 2023, 218, 113153, doi:10.1016/j.measurement.2023.113153.
[4] Abdeljaber, O.; Avci, O.; Kiranyaz, M.S.; Boashash, B.; Sodano, H.; Inman, D.J. 1-D CNNs for Structural Damage Detection: Verification on a Structural Health Monitoring Benchmark Data. Neurocomputing 2018, 275, 1308–1317, doi:10.1016/j.neucom.2017.09.069.
[5] Zhang, J.-K.; Yan, W.; Cui, D.-M. Concrete Condition Assessment Using Impact-Echo Method and Extreme Learning Machines. Sensors 2016, 16, 447, doi:10.3390/s16040447.
[6] Lin, S.; Meng, L.; Zhao, G.; Zhang, J.; Xin, J.; Cheng, Y.; Cheng, S.; Zhai, C. Automatic Elimination of Invalid Impact-Echo Signals for Detecting Delamination in Concrete Bridge Decks Based on Deep Learning. Dev. Built Environ. 2024, 19, 100521, doi:10.1016/j.dibe.2024.100521.

Round 2
Reviewer 1 Report
Comments and Suggestions for Authors
The authors well addressed my concerns. The manuscript can be published in the current version.
Reviewer 2 Report
Comments and Suggestions for Authors
The revised manuscript has been significantly improved and can therefore be accepted for publication.